# A Method to Reduce the Occurrence of Egg Translucency and Its Effect on Bacterial Invasion

**DOI:** 10.3390/foods12132538

**Published:** 2023-06-29

**Authors:** Xuefeng Shi, Qianni Liang, Enling Wang, Caiyun Jiang, Lingsen Zeng, Ruochen Chen, Junying Li, Guiyun Xu, Jiangxia Zheng

**Affiliations:** 1College of Animal Science and Technology, China Agricultural University, Beijing 100193, China; xuefeng.shi@cau.edu.cn (X.S.); lqn17603727160@163.com (Q.L.); cyjiang@cau.edu.cn (C.J.); s20203040561@cau.edu.cn (L.Z.); tianxietianyu@163.com (R.C.); lijunying@cau.edu.cn (J.L.); ncppt@cau.edu.cn (G.X.); 2College of Animal Science and Technology, Huazhong Agricultural University, Wuhan 430070, China; 13466646968@139.com

**Keywords:** eggshell translucency, MDCP, chronic heat stress, eggshell ultrastructure, egg quality, *E. coli*

## Abstract

Translucent egg consumption is low due to consumer acceptance and quality concerns, which is a problem that egg producers need to address. This study was performed to evaluate the reasons for the high occurrence of egg translucency in summer, as well as whether the addition of mono-dicalcium phosphate (MDCP) to the diet can relieve eggshell translucency and whether eggshell translucency is associated with the risk of bacterial invasion. A total of 72 laying hens that were 36 weeks old were randomly divided into control (CON) and MDCP groups and fed in the same environment. Results showed that the number of translucent eggs increases in July and August as the temperature and humidity increase. Compared with the CON group, in July, August, and October, the translucent egg grade (TEG) of the MDCP group was lower than that of the CON group (*p* < 0.05). TEG was correlated with mastoid space height (MSH), width (MSW), and area (MSA) (correlation coefficients 0.63, 0.59, and 0.68, respectively, *p* ≤ 0.05). There was no significant difference in the invasion rate of *E. coli* between translucent and non-translucent egg groups (47.2% vs. 39.33%), and translucent area and non-translucent area (13.49% vs. 15.08%). In conclusion, our results show that dietary MDCP may alleviate eggshell translucency and that eggshell translucency would not increase the probability of *E. coli* cross-shell penetration rate.

## 1. Introduction

Eggs are a rich source of protein and provide a unique, well-balanced food source with high nutritional value for people of all ages [1]. When consumers choose shell eggs, they are influenced by the outer quality of the eggs [2]. Translucency is a phenomenon in which the eggshell surface appears dark under sunlight because of the uneven distribution of water in the eggshell surface, which seriously affects the egg’s appearance and diminishes its economic value [3]. As consumer awareness of egg quality has increased, translucent eggs have become a focus for improvement in laying hen egg production [4].

Eggshell translucency is caused by structural problems in the eggshell [5]. It is currently believed that changes in the structure of the eggshell papilla layer and the eggshell membrane result in the accumulation of water in the eggshell [3]. Summer is a period of high incidence of egg translucency. In addition, the breed and age of the hen, as well as feed and environmental factors (such as temperature and humidity) are known to affect eggshell translucency [6,7,8]. The external temperature and humidity are important environmental factors affecting poultry production [9,10]. Lin et al. [11] showed that the optimal ambient temperature for laying hens is 19–22 °C, with higher temperatures likely to induce heat stress. Heat stress can affect the production performance of hens [9]. The laying hens are subjected to long-term heat stress, which not only affects the egg production rate and eggshell quality but also affects the health of the laying hens in severe cases, which in turn increases the translucency of the eggs [12,13,14]. Therefore, alleviating the impact of heat stress on poultry is an important task in maintaining their health and productivity.

Chousalkar et al. [15] reported that changes in the mastoid and mastoid core layers in the eggshell formation stage increase the occurrence rate of eggshell cracks, allowing Salmonella and *Escherichia coli* (*E. coli*) to cross easily via eggshell translucencies. However, Shi et al. [16] and Chen et al. [17] showed no correlation between eggshell translucency and eggshell thickness. Furthermore, Chen et al. [17] revealed that the quality of the cuticle and thickness of the eggshell are the main factors that determine the cross-shell invasion of bacteria, which suggests that the relationship between eggshell translucency and cross-shell bacterial contamination is not inevitable. Whether eggshell translucency increases the probability of bacterial cross-shell contamination thus remains to be confirmed.

Feed phosphate is a mineral additive that plays a role in animal metabolism and can improve animal growth and production performance [18]. Currently, feed phosphate products mainly include dicalcium phosphate (DCP) [CaHPO_4_·2H_2_O], monocalcium phosphate (MCP) [Ca(H_2_PO_4_)2·H_2_O], and mono-dicalcium phosphate (MDCP) [CaHPO_4_·2H_2_O·Ca (H_2_PO_4_)2·H_2_O] [19]. DCP is the most widely used phosphate compound, but the phosphorus in DCP is barely soluble in water, and the absorption rate of its effective ingredients by poultry and swine is low (less than 60%) [20]. The remaining phosphorus and calcium elements are discharged in feces, causing pollution of groundwater resources, soil, etc. MDCP is a eutectic combination of calcium dihydrogen phosphate and DCP, with an absorption rate of its effective components (>75%), which is higher than that of DCP [21]. The amount of residual phosphorus in feces is lower, and it is more environmentally friendly. Together with national requirements for environmental protection and consumers’ high expectations regarding the quality of livestock products, the use of MDCP as a phosphorus source has many advantages [22]. In addition, calcium and phosphorus are indispensable elements in the process of eggshell formation [23,24]. However, while there have been many studies examining the effect of MDCP on the growth performance of broilers and ducks, there are only a few studies that have focused on the effect of MDCP on eggshell translucency and eggshell quality [25,26,27].

In this study, we investigated the effect of adding MDCP to the diet on eggshell translucency in Dwarf pink-shell laying hens under heat stress. We further explored whether the eggshell translucency affects the cross-eggshell invasion of *E. coli* to determine whether the appearance quality of translucent eggs is linked to their safety.

## 2. Materials and Methods

### 2.1. Animal Ethics

All experiments were performed in compliance with the Guidelines for Experimental Animals established by the Animal Care and Use Committee of China Agricultural University (permit number: AW22119102-1-1). All animal experiments also comply with the ARRIVE guidelines.

### 2.2. Experimental Materials and Feeding Management

The experiment was conducted at the Zhuozhou Poultry Testing Center of China Agricultural University (Zhuozhou, China). The pre-feeding period took place in the last week of June and the experimental period was from 1 July to 31 October. A total of 72 Dwarf pink-shell laying hens (36 weeks, 2.00 ± 0.10 kg) were used. These laying hens were randomly divided into two groups: CON and MDCP. All hens were raised in the same hen house on the same diet. The composition and nutrition of the basic diets are shown in Table 1. We added 8000 mg of DCP to the CON group and 5900 mg of MDCP to the MDCP group per kilogram of basal diet to ensure that the phosphorus content in the two groups was consistent. Each experimental group had 12 replicate groups, with each replicate group comprising three hens. During the experiment, we recorded the average daily feed consumption per laying hen, egg production, broken egg, and death rates of the hens. We also recorded the temperature and relative humidity of the chicken house and egg storage room at 2:00 p.m. daily.

### 2.3. Sample Collection

During the entire experimental period, eggs were collected from all experimental groups at the end of each month. Egg quality was determined on the day of collection. Eggs were stored at 25 °C and 65% relative humidity (RH) for three days, after which eggs were graded for translucency. After the egg whites and yolks were removed, the eggshells were washed with distilled water and dried at room temperature overnight, prior to analyses of eggshell ultrastructure. In addition, the blood of the hens was collected every two months. Thirty hens were isolated from each group, and 1 mL of blood was collected from the wing vein, centrifuged after standing and stored at −80 °C. Serum was used to determine the blood biochemical indexes.

### 2.4. Serum Biochemical Indexes

Blood biochemical indices were measured every two months, including triiodothyronine (T3), thyroxine (T4), and corticosterone (CORT). An ELISA kit (Shanghai Yuan Mu Biotechnology Co., Ltd., Shanghai, China) was used to determine the concentrations of T3, T4, and CORT in the serum. All analyses were repeated, and results with deviations exceeding 5% within the sample were rejected.

### 2.5. Grading of Translucent Eggs

After 3 days of storage, based on the size and density of the translucent eggs, we selected six eggs as reference samples for grading from 0 to 5 (Figure 1). Grade 0 was excellent, with only a few slightly bright spots under LED light and no black spots on the eggshell surface under natural light. At each time point, 30 eggs from each group were measured for translucency. The grading process was performed in a dark room. First, under dark conditions, we used LED lights (BULL, Beijing, China) to illuminate all the eggs and grade them according to the reference samples. To reduce random error, three members scored the eggs individually, and the score was determined to be valid only when at least two experimenters provided consistent scores. Otherwise, eggs were reevaluated and regraded.

### 2.6. Eggshell Ultrastructure

Analyses of eggshell ultrastructure were performed on 30 eggs from each group every 2 months. Eggshell pieces representing a longitudinal section and the outer surface of the eggshell were each fixed to a metal platform, and the gap between the platform and the eggshell was filled with metal glue, sprayed with gold powder and allowed to dry for 15 min, and then examined by scanning electronic microscope (SU3800, HITACHI, Tokyo, Japan). Image J software was used to measure the mastoid space (MS), width (MSW), height (MSH), area (MSA), and shell membrane thickness (SMT) (Figure 2).

### 2.7. Egg Quality

Egg quality was measured once a month over the 4-month period of the experiment. Thirty eggs were collected from each group to determine egg quality by measuring egg weight (EW), eggshell strength (ESS), Haugh unit (HU), eggshell thickness (EST), and yolk color (YC). The EW, HU, and YC of the eggs were measured using an EMT-5200 multi-function egg tester (Robotmation, Tokyo, Japan). ESS was measured using a Model-II eggshell strength tester (Robotmation, Tokyo, Japan). EST was performed using a micrometer screw gauge to measure three eggshell positions—bottom, middle, and top—and the average value was calculated.

### 2.8. Determination of E. coli Invasion Rate

We followed protocols previously described for cuticle quality to determine the effect of eggshell translucency on the invasion rate of *E. coli* [17]. The eggs were divided into translucent (the TEG of eggs is 3) and non-translucent (the TEG of eggs are 0) egg groups, and the translucent eggs were further subdivided into translucent and non-translucent. Strains of *E. coli* containing the plasmid pGLO (Bio-Rad Laboratories, Ontario, CA, USA) were incubated overnight at 37 °C with shaking in agarose nutrient solution medium (10 g tryptone, 5 g yeast extract, and 10 g sodium chloride; Shanghai, China) containing 100 μg/mL ampicillin Na and 5 mM L-arabinose (Sigma-Aldrich, Shanghai, China) at 37 °C. The experimental culture was inoculated with culture at a dilution of 1:50 and placed into an ice bath to chill when the culture reached an OD600 of approximately 0.4. Eggs were gently wiped with 75% alcohol to sterilize them, placed in a clean sterile plastic-sealed egg box, and incubated for 3 h at 37 °C before inoculation. The eggs were then completely immersed in the chilled *E. coli* culture and incubated in an ice bath for 15 min. The inoculated eggs were then dried for 10 min on a bench sterilized with 75% ethanol and 10% bleach. Finally, the eggs were placed in clean sterile plastic bags and incubated at 37 °C for 24 h. The eggs were then drained and cut in half along the long axis using a cutter (DREMEL 4300, BOSCH, Stuttgart, Germany). The eggshells were wiped clean with clean paper and examined under an ultraviolet lamp for luminescent spots on the inside of the eggshell, indicating invasion by pGLO-expressing *E. coli*.

### 2.9. Statistical Analyses

Differences between the experimental and control groups were analyzed by the general linear model using SPSS software (version 23.0, SPSS Inc., Chicago, IL, USA) and modeled as follows:Y_ij_ = μ + a_i_ + b_ij_
where μ is the overall average, a_i_ is the effect of groups, and b_ij_ is the residual error. The significance of differences between groups was analyzed by Duncan’s multiple comparison tests.

The basic descriptive statistical results are shown as the mean with standard deviation (mean ± SEM). The coefficient of variation (CV) was used to compare the dispersion of the different groups of data with regard to eggshell structure. ImageJ 1.34s software (NIH, Bethesda) was used to measure the microstructure of the eggshells. The chi-square test was used to compare *E. coli* invasion rates. GraphPad Prism 7.0 (GraphPad Prism Software Inc., San Diego, CA, USA) software was used to plot the experimental results.

## 3. Results

### 3.1. State of Chronic Heat Stress in Different Months

With reference to the heat stress index (HIS) proposed by Beshir and Ramsey [28] and considering the influence of temperature and relative humidity of the environment in the chicken house, we divided heat stress into critical, danger, warning, and comfort areas (Figure 3). The temperature and humidity readings in July and August were almost entirely distributed in the warning and danger areas, while those in September and October were almost entirely distributed in the comfort area. This shows that during July and August, the hens were subjected to chronic heat stress. During the experiment, there were no significant differences among the groups in average daily feed consumption, laying egg rate, broken egg rate, and death rate of hens.

### 3.2. Effect of Adding MDCP on Blood Heat Stress Indexes

The effects of adding MDCP on blood biochemical indicators are shown in Table 2. In August and October, T3 levels in the MDCP group were higher than those in the CON group (*p* < 0.05). The levels of T4 in the CON and MDCP groups in August and October were higher than those in June (*p* < 0.05). In August, T4 levels in the MDCP group were higher compared with the CON group (*p* < 0.05). In the same month, there was no significant difference in CORT levels between the CON and MDCP groups. However, in the MDCP group, CORT levels in August and October were higher than those in June (*p* < 0.05).

### 3.3. Effect of Adding MDCP on Eggshell Translucency

The effects of adding MDCP on eggshell translucency are shown in Figure 4. There was no significant difference in translucent egg grades between the CON group and the MDCP group when feeding was started in June. However, in July, August, and October, the translucent egg grades of the MDCP group were lower than those of the CON group (*p* < 0.05). In September, the translucent egg grades of the MDCP group were lower than those of the CON group, but there was no significant difference.

### 3.4. Correlation between Eggshell Translucency and Structure

The correlation between the translucent eggs grades (TEG) and eggshell structure (MSH, MSW, MSA, and SMT) is shown in Figure 5. The correlations between TEG and MSH, MSW, MSA, and SMT were 0.63, 0.59, 0.68, and 0.22, respectively. TEG is highly correlated with MSH, MSW, and MSA (*p* < 0.05).

### 3.5. Effect of Adding MDCP on Eggshell Structure

The effect of adding MDCP on the eggshell structure is shown in Table 3. Within the groups, the MSH, MSW, MSA, and SMT values exhibited significant differences between different months (*p* < 0.05). In October, MSH, MSW, and MSA all showed significant differences (*p* < 0.05) between the CON and MDCP groups. Interestingly, with the increase in months, the CV value of the MDCP group gradually decreased, while the CV value of the CON group did not show such a regularity. SMT did not show significant differences and regularity between the two groups. We, therefore, reasoned that dietary supplementation with MDCP might reduce egg translucency by making the eggshell structure more homogeneous and reducing variability in the mastoid layer.

### 3.6. Effects of Adding MDCP on Egg Quality

The effects of adding MDCP on egg quality are shown in Table 4. The EW of the two groups showed an overall upward trend with increasing age. In the same month, there was no significant difference in egg quality between the CON group and the MDCP group.

### 3.7. Effect of Eggshell Translucency on Bacterial Invasion Rate

The effect of eggshell translucency on the bacterial invasion rate is shown in Figure 6. When the inner surface of the eggshells was observed under ultraviolet light, the presence of luminescent spots indicated *E. coli* invasion (Figure 6A). The invasion rate of *E. coli* in the translucent egg group was higher than that in the non-translucent egg group (47.22% vs. 39.33%, *p* = 0.6873), but this difference was not significant, indicating that eggshell translucency had no effect on the cross-shell invasion of *E. coli* (Figure 6B). Interestingly, the invasion rate of *E. coli* in non-translucent areas was higher than that in translucent areas of the same egg (15.08% vs. 13.49%, *p* = 0.8571); however, this difference was also not significant, which further indicated that the eggshell translucency had no effect on the cross-shell invasion of *E. coli* (Figure 6B).

## 4. Discussion

The optimum ambient temperature for poultry production is 19–22 °C [11]. At temperatures that are higher than this range, poultry experience heat stress [29], which has been shown to reduce feed intake and conversion efficiency [30]. Heat stress affects not only the daily weight gain of meat poultry but also the egg production rate, fertilization rate, and reproductive performance of egg-laying and breeding poultry [31,32]. It also impairs the normal physiological functions of poultry, causing great harm to their health [33]. Galarza-Seeber et al. [34] noted that heat stress changes the intestinal morphology and connections of the epithelial junctions, adversely affecting intestinal permeability and integrity. Zhang et al. [35] showed that the increase in intestinal permeability caused by heat stress allowed intestinal pathogenic bacteria such as Salmonella, Clostridium, and *E. coli* to penetrate the gastrointestinal tract, which eventually led to a decline in the immune performance of hens. In addition, heat stress in poultry causes oxidative stress due to the excessive production of reactive oxygen species, eventually leading to oxidative damage [36]. Heat stress also affects food safety and reduces the quality of poultry meat and eggs, thus affecting human nutrition and health [29,37]. In our study, during the months of July and August (peak summer in the northern hemisphere), the hens were in a state of heat stress, which inevitably affected their production performance.

Under high-temperature conditions, birds change their behavior and physiological stability to help regulate and thus lower their body temperature [11,38]. In general, hens have similar responses to heat stress, but there are individual differences in the intensity and duration of their responses [39]. For example, body temperature and metabolic activity are regulated by T3 and T4, and the balance between them, and high-temperature environments can lead to higher concentrations of CORT in serum [40,41].

Previous studies have reported that under heat stress, the concentration of T3 continues to decrease, but the concentration of T4 either increases, decreases, or remains unchanged [41]. In this study, the concentration of T3 decreased under heat stress, whereas the concentration change of T4 was not as consistent as that of T3. In August, the concentrations of T3 and T4 in the F2 group were higher than those in the CON group (*p* < 0.05). We speculate that this is because the addition of MDCP to the diet reduces the heat stress response of these hens, which in turn affects the distribution of T3 and T4 concentrations in the serum. Because of the high temperature, the T3 and T4 concentrations decreased and increased, respectively, which is consistent with results from other studies. The concentration of T4 in the serum of laying hens tends to increase after 6 h of high-temperature exposure [38]. Buyse et al. [39] reported that T3 plays a crucial role in the regulation of hen temperature and that serum T3 levels are positively correlated with heat generation.

The concentration of CORT in the serum increased under high temperatures, which is also consistent with findings from other studies [42,43]. CORT can regulate some performance indexes and immune parameters in chickens [11]. Studies have shown that broilers injected with CORT gain less weight [44]. In addition, CORT added to the diet or drinking water can reduce the growth performance of chickens [42,45]. Quinteiro-Filho et al. [46] showed that heat stress can improve the level of CORT in serum by activating the hypothalamic neural network.

Eggshells are mainly composed of inorganic (95–97%) and organic (3–5%) matter, in which the inorganic matter mainly consists of 95% calcium carbonate and a small amount of magnesium carbonate, calcium carbonate, and magnesium phosphate [47]. Calcium and phosphorus are the two main elements in eggshell composition; therefore, adding calcium and phosphorus to the diet is particularly important for eggshell formation [48]. In August, the number of translucent eggs increased when hens experienced chronic heat stress. Interestingly, the translucent egg score of the MDCP group was lower than that of the CON group during July and August, indicating that the addition of MDCP to the diet can alleviate eggshell translucency under chronic heat stress conditions (*p* < 0.05). Nie et al. [48] reported that a 0.4% non-plant phosphate diet reduced the incidence of translucent eggs, suggesting that calcium and phosphorus metabolism is related to egg translucency.

The eggshell is primarily composed of calcium carbonate crystals arranged in a columnar shape [47]. Nie et al. [49] showed that the calcium content in the outer layer of the eggshell and the phosphorus content in the eggshell membrane of translucent eggs were significantly lower than those of non-translucent eggs. In the case of stress or disease, these calcium carbonate crystals exhibit a disordered arrangement that allows water to gather in the gaps between the crystals, which become translucent spots after drying [50]. Studies have shown that the formation of translucent eggs is related to the chemical composition of the eggshell structure and changes in the eggshell membrane, such as thinning of the effective layer and increases in papillary space [51]. Zhao et al. [52] reported that the addition of 25 hydroxyvitamin D and essential oil complex to the diet can significantly reduce the incidence of translucent eggs by increasing the effective thickness of eggshells and decreasing the thickness of the mastoid. Zhang et al. [53] reported that adding manganese to the diet can increase the density of eggshell mammillary knobs to reduce the occurrence of translucent eggs. Our research showed that the addition of MDCP reduced the variation in eggshell microstructure and made the eggshell microstructure more uniform. We speculate that it was precisely because of this homogeneity that the incidence of translucent eggs decreased.

Fu et al. [54] showed that an increase in the mastoid space in the eggshell provides more water storage space and increases the incidence of translucent eggs. Chousalkar et al. [15] reported that the larger mastoid space in the translucent area of the eggshell may be related to variations in the early mastoid and mastoid cores. In their study, Talbot and Tyler [50] suggest that the porosity of the translucent area is high, which results in the speed of the content penetrating the eggshell being greater than the speed of eggshell water volatilization such that water gathers in the eggshell and causes translucent eggs; however, the areas without translucency demonstrate low porosity. In our correlation study, our test results showed a significant positive correlation between TEG and MSH, MSA, and MSW (*p* < 0.05). However, in our experimental results, the correlation between TEG and SMT was low.

Studies have also shown that the longitudinal tensile breaking force and thickness of the eggshell membrane of translucent eggs is lower than that of normal eggs, indicating that the membrane is more likely to break in translucent eggs, thus causing the water inside the egg to accumulate in the eggshell through the eggshell membrane to form translucent eggs [3]. Oil can protect the components of eggshell membranes, improve the oxidation resistance of eggshell membranes, and reduce the permeability of water [55]. Studies have shown that adding 1.85% mixed oil to the diet can significantly reduce the incidence of translucent eggs [56].

A high-temperature environment causes stress reactions in the body and affects the production performance of hens, which goes on to affect egg quality [57]. In this study, we found no significant differences in egg quality between the groups. Negoi et al. [58] found that shell weight increases with calcium intake during egg laying. Additionally, Tischler et al. [59] found that a low phosphorus diet significantly increased egg weight and reduced eggshell strength in laying hens. Under constant heat stress, HU and protein height do not change significantly [60,61]. In an experiment conducted by Franco Jiminez et al. [60], heat stress reduced total egg weight and protein weight. Our research showed that with the increase in week age, egg weight showed a significant difference (*p* < 0.05), but there was no significant difference between the groups in the same month.

Translucent eggs can reduce consumers’ willingness to buy such eggs due to their appearance; they are also considered to increase egg quality safety risks, affect egg consumption, and eventually cause economic losses to the laying hen industry [3]. Few studies have been conducted on eggshell translucency and bacterial cross-shell contamination. Chousalkar et al. [15] showed that eggshell translucency is related to changes in the mastoid layer and mastoid cores during the early phases of eggshell formation, which may lead to cracks in the eggshell, thus increasing bacterial penetration. In contrast, Fu et al. [54] and Wang et al. [3] reported that the formation of translucent eggs is mainly caused by the accumulation of moisture in the eggshell mastoid layer and eggshell membrane. However, when we compared translucent eggs with non-translucent eggs, we did not find that eggshell translucency was related to penetration by *E. coli*.

Studies have shown that cuticle, eggshell thickness, and eggshell pore diameter are the main factors that determine bacterial invasion across the shell [17,62,63]. The cuticle is the outermost layer of the eggshell—when the pores on the eggshell surface are evenly covered, the cuticle prevents bacteria from invading the inside of the egg through the eggshell pore [17,64]. The thickness of the eggshell is a key factor that affects the length of the cross-shell invasion path of the bacteria. The thinner the eggshell, the easier it is to contaminate the egg [62]. The eggshell pore diameter also limits the penetration ability of bacteria by size exclusion [65]. Chen et al. [17] reported that the antibacterial efficiency of eggshells is determined by the thicknesses of the cuticle and eggshell. When the thickness of the eggshell was greater than 340 μm and the α value of the cuticle was greater than 27.5%, the antibacterial efficiency of the eggshell was as high as 98%. Shi et al. (2018) reported that the thicknesses of the cuticle and eggshell had no significant correlation with eggshell translucency. To eliminate the influence of cuticle, eggshell thickness, and eggshell pore diameter on the experimental results, we also compared translucent and non-translucent areas with regard to the cross-shell invasion rate of *E. coli* in the same egg. Our results showed that there was no significant difference in the *E. coli* invasion rate between the translucent and non-translucent areas in the same egg, which provides further evidence that eggshell translucency has no effect on the cross-shell invasion of *E. coli*.

Therefore, our research shows that the addition of MDCP to the diet is valuable and should be further developed. Compared with other methods, it is a relatively simple and low-cost method to reduce the incidence of translucent eggs. Although translucent eggshells will not increase the rate of bacterial invasion, their unique appearance will affect consumers’ desire to buy. From our research results, we can conclude that by adding MDCP to the diet, the occurrence rate of translucent eggs can be reduced, making it easier for consumers to accept the product and provide support for the development of the industry.

## 5. Conclusions

In conclusion, heat stress can cause changes in eggshell microstructure and increase the incidence of translucent eggs. Dietary MDCP supplementation can improve the effect of chronic heat stress on translucent eggs, which likely occurs through its ability to enhance the eggshell ultrastructure. Furthermore, our study has demonstrated for the first time that eggshell translucency is not related to cross-shell invasion of *E. coli*, which will help to further our understanding of the relationship between eggshell translucency and egg quality safety.

## Figures and Tables

**Figure 1 foods-12-02538-f001:**
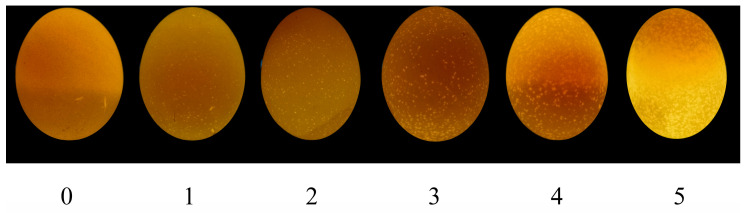
The six grading reference samples used for the scoring method are assigned 0, 1, 2, 3, 4, and 5 points, respectively, from left to right. Grade 0 is the best, with no bright spots under the LED light. Grade 1 is fine, with a few slightly bright spots under the LED light. Grade 2 is good, with many slightly bright translucent spots distributed over the shell under the LED light. Grade 3 is moderate, with many translucent spots distributed over the shell under the LED light. Grade 4 is severe, with dense small and large bright spots on the surface under the LED light. Grade 5 is extremely severe, with numerous bright spots on the surface under the LED light.

**Figure 2 foods-12-02538-f002:**
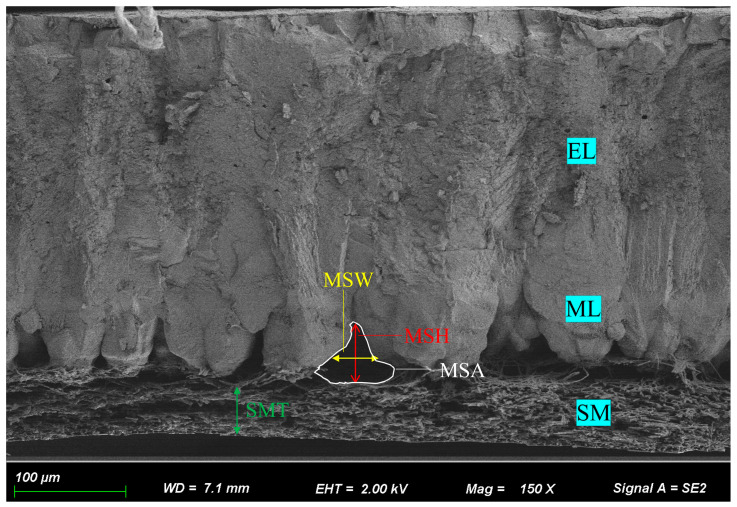
Scanning electron microscope images of vertical profiles of laying hens’ eggshells. EL, effective layer; ML, mastoid layer; SM, shell membrane; MSH, mastoid space height; MSW, mastoid space width; MSA, mastoid space area; SMT, shell membrane thickness.

**Figure 3 foods-12-02538-f003:**
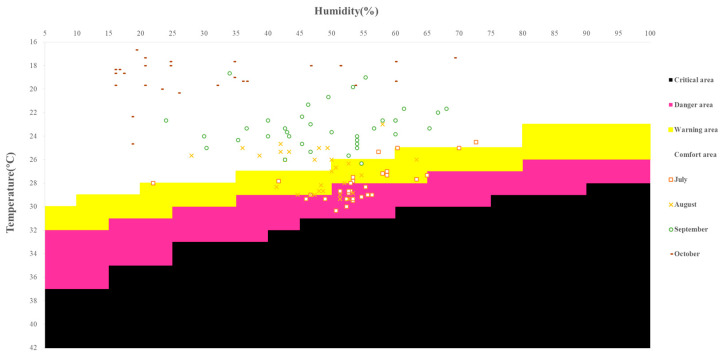
Daily environmental temperature and humidity levels distribution within different tolerance areas.

**Figure 4 foods-12-02538-f004:**
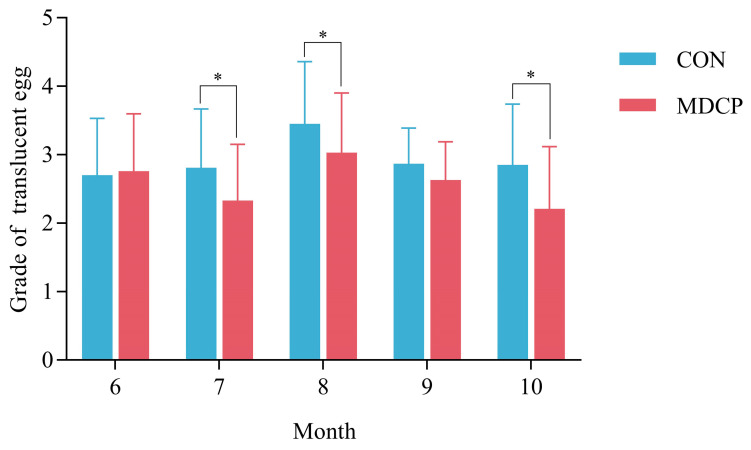
Comparison of the translucent egg scores between groups in different months. (* *p* < 0.05).

**Figure 5 foods-12-02538-f005:**
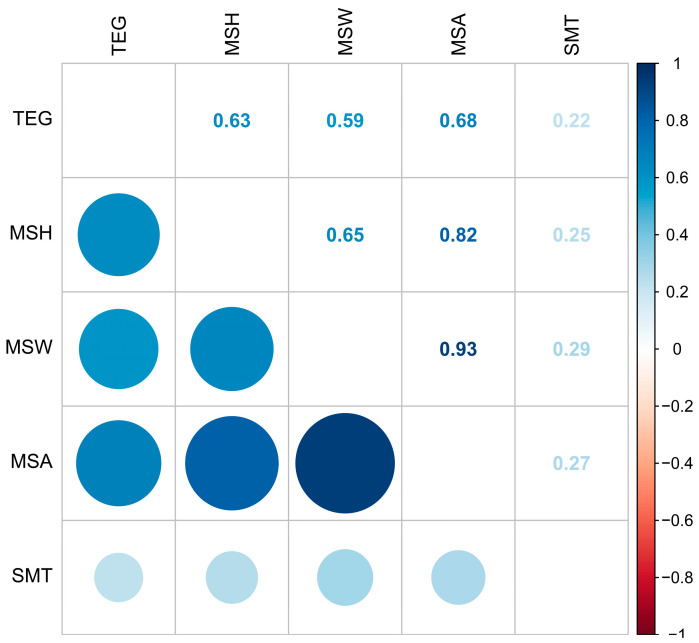
The correlation percentage between different eggshell parameters. Color and size represent correlation; both darker color and larger size mean a greater correlation. TEG, translucent egg grade; EL, effective layer; ML, mastoid layer; SM, shell membrane; MSH, mastoid space height; MSW, mastoid space width; MSA, mastoid space area; SMT, shell membrane thickness.

**Figure 6 foods-12-02538-f006:**
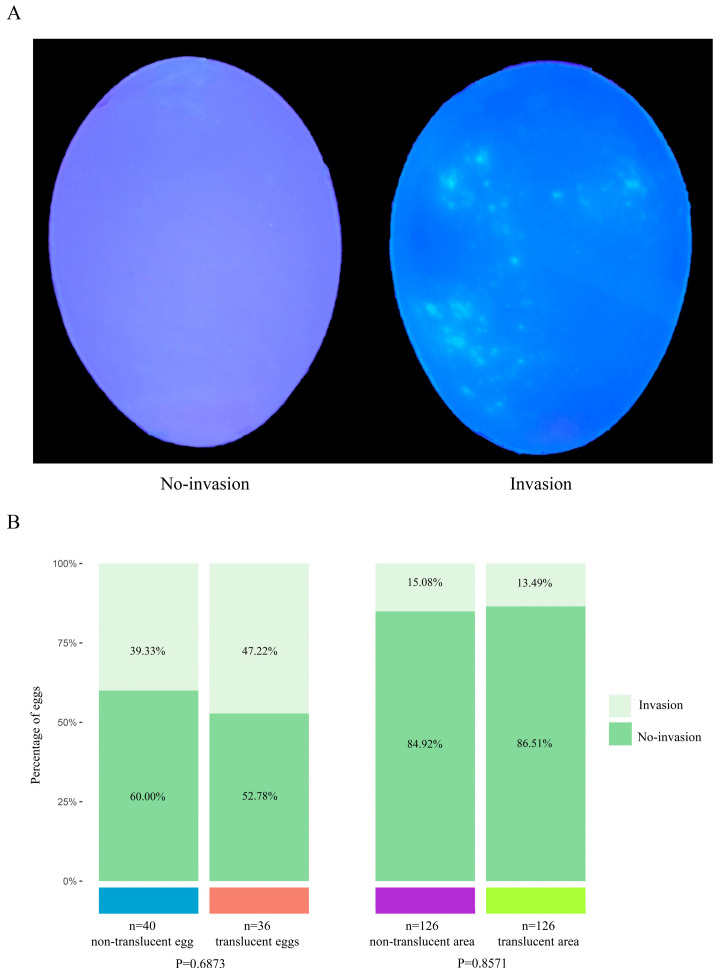
Effect of eggshell translucency on the cross-shell penetration rate of *E. coli.* (**A**) Egg with no *E. coli* invasion compared with the egg with the invasion of *E. coli*. (**B**) Comparison of the influence of the translucent egg group and non-translucent egg group, and translucent area and non-translucent area on the invasion rate of *E. coli*.

**Table 1 foods-12-02538-t001:** Composition and nutrient levels of the basal diet (%, air-dry basis).

Items		Items	
Ingredients (%)		Nutrient levels ^3^	
Corn	62.50	Metabolic energy (MJ/Kg)	11.42
Soybean meal	26.20	Total methionine	0.42
Soy oil	1.00	Crude protein	16.04
Limestone	9.58	Total lysine	0.81
Sodium chloride	0.30	Total threonine	0.60
Choline chloride	0.10	Ca	3.64
Vitamin premix ^1^	0.02	Available phosphorus	0.35
Trace mineral premix ^2^	0.30		

^1^ Provided per kg of feed: Vitamin A, 14,000 IU; Vitamin D_3_, 4500 IU; Vitamin E, 70 mg; Vitamin K3, 3.5 mg; Vitamin B_1_, 3.0 mg; Vitamin B_2_, 6.9 mg; Vitamin B_6_, 4.6 mg; Vitamin B_12_, 0.025 mg; Pantothenic acid, 10 mg; Niacin 35 mg; Folic acid, 1 mg; Vegetable acid 100 mg; ^2^ Provided per kg of feed: Fe, 60 mg; Cu, 15 mg; Mn, 120 mg; Zn, 110 mg; I, 1.2 mg. ^3^ Calculated values.

**Table 2 foods-12-02538-t002:** Comparison of T3, T4, and CORT of different groups at different times.

Item	Group	Month			*p*-Value
		6	8	10	
T3 (nmol/L)	CON	5.19 ± 0.28	4.93 ± 0.25 ^b^	5.48 ± 0.25 ^b^	0.37
	MDCP	5.70 ± 0.31 ^B^	7.05 ± 0.50 ^A,a^	6.55 ± 0.33 ^AB,a^	0.03
	*p*-value	0.23	<0.01	0.01	
T4 (nmol/L)	CON	60.66 ± 1.81 ^A^	55.49 ± 2.23 ^AB,b^	51.71 ± 1.92 ^B^	<0.01
	MDCP	61.66 ± 2.47 ^AB^	65.74 ± 3.23 ^A,a^	55.06 ± 2.36 ^B^	<0.01
	*p*-value	0.75	0.01	0.28	
CORT (ng/mL)	CON	82.53 ± 3.26	90.03 ± 2.84	81.37 ± 3.25	0.27
	MDCP	83.63 ± 2.43 ^B^	93.44 ± 3.84 ^A^	84.82 ± 3.42 ^AB^	<0.01
	*p*-value	0.79	0.48	0.47	

^A,B^ means within a row (months) that do not share a common superscript differ significantly (*p* < 0.05). ^a,b^ means within a column (Groups) by T3, T4 and CORT that do not share a common superscript differ significantly (*p* < 0.05). Triiodothyronine, T3; thyroxine, T4; corticosterone, CORT.

**Table 3 foods-12-02538-t003:** Comparison of MSH, MSW, and MSA of different groups at different times.

Items	Month	CON		MDCP		*p*-Value
		Mean ± SEM	CV/%	Mean ± SEM	CV/%	
MSH (μm)	6	29.58 ± 0.97 ^a^	17.95	29.00 ± 1.07 ^a^	20.17	0.67
	8	26.06 ± 0.64 ^b^	13.39	25.08 ± 0.53 ^b^	11.48	0.98
	10	21.70 ± 0.57 ^B,c^	14.29	25.24 ± 0.44 ^A,b^	9.55	<0.01
	*p*-value	<0.01		<0.01		
MSW (μm)	6	27.31 ± 0.74 ^a^	14.90	28.33 ± 0.89	17.26	0.34
	8	26.74 ± 0.72 ^a^	14.85	25.98 ± 0.75	15.86	0.43
	10	23.48 ± 0.86 ^B,b^	20.06	27.30 ± 0.75 ^A^	15.02	<0.01
	*p*-value	<0.01		0.12		
MSA (μm^2^)	6	533.88 ± 30.07 ^a^	30.85	533.74 ± 33.57	34.45	0.10
	8	466.67 ± 16.05 ^b^	18.84	461.44 ± 13.67	16.22	0.79
	10	375.74 ± 17.64 ^B,c^	25.72	471.45 ± 12.55 ^A^	14.58	<0.01
	*p*-value	<0.01		0.06		
SMT (μm)	6	60.92 ± 1.47 ^a^	13.30	65.10 ± 1.57 ^a^	13.20	0.06
	8	54.82 ± 1.64 ^b^	16.36	54.58 ± 1.15 ^b^	20.67	0.91
	10	56.10 ± 2.04 ^ab^	18.89	60.04 ± 1.65 ^ab^	23.31	0.14
	*p*-value	0.04		0.02		

^A,B^ means within a row (Groups) that do not share a common superscript differ significantly (*p* < 0.05). ^a,b,c^ means within a column (Months) of the common items that do not share a common superscript differ significantly (*p* < 0.05). MSH, mastoid space height; MSW, mastoid space width; MSA, mastoid space area; SMT, shell membrane thickness.

**Table 4 foods-12-02538-t004:** Comparison of the egg quality of different groups at different times.

Items	Month	Group		*p*-Value
		CON	MDCP	
EW (g)	6	48.62 ± 0.64 ^b^	47.92 ± 0.57 ^c^	0.42
	7	51.88 ± 0.80 ^a^	50.60 ± 0.62 ^b^	0.21
	8	50.91 ± 0.66 ^a^	50.89 ± 0.81 ^b^	0.99
	9	51.92 ± 0.51 ^a^	52.67 ± 0.52 ^a^	0.31
	10	52.08 ± 0.73 ^a^	53.57 ± 0.50 ^a^	0.10
	*p*-value	<0.01	0.02	
ESS (kg/cm^2^)	6	3.15 ± 0.10	3.03 ± 0.07	0.34
	7	2.98 ± 0.09	3.05 ± 0.13	0.67
	8	2.97 ± 0.10	2.92 ± 0.08	0.69
	9	2.92 ± 0.10	3.03 ± 0.09	0.43
	10	3.04 ± 0.08	2.98 ± 0.09	0.64
	*p*-value	0.39	0.70	
HU	6	69.37 ± 2.00 ^bc^	64.75 ± 1.40 ^b^	0.06
	7	66.19 ± 1.78 ^c^	69.96 ± 2.13 ^ab^	0.18
	8	64.91 ± 1.98 ^c^	66.83 ± 2.05 ^ab^	0.50
	9	72.60 ± 2.00 ^ab^	71.86 ± 1.68 ^a^	0.78
	10	76.94 ± 2.01 ^a^	72.20 ± 1.65 ^a^	0.07
	*p*-value	<0.01	<0.01	
EST (μm)	6	349.56 ± 6.14	340.77 ± 6.07	0.31
	7	332.31 ± 5.42	331.74 ± 6.21	0.95
	8	342.47 ± 6.22	339.73 ± 5.86	0.73
	9	343.68 ± 5.48	343.38 ± 5.40	0.97
	10	345.82 ± 6.08	334.85 ± 6.04	0.21
	*p*-value	0.76	0.97	
YC	6	6.38 ± 0.12 ^b^	6.37 ± 0.13 ^b^	0.95
	7	6.48 ± 0.10 ^b^	6.47 ± 0.09 ^b^	0.94
	8	6.30 ± 0.16 ^b^	6.17 ± 0.12 ^b^	0.51
	9	5.30 ± 0.14 ^c^	5.37 ± 0.16 ^c^	0.75
	10	7.20 ± 0.13 ^a^	7.39 ± 0.09 ^a^	0.24
	*p*-value	0.05	0.10	

^a,b,c,d^ means within a column (Months) of the common items that do not share a common superscript differ significantly (*p* < 0.05). EW, egg weight; HU, Haugh units; YC, yolk color; ESI, egg shape index; ESS, eggshell strength; EST, eggshell thickness.

## Data Availability

The data that support the findings of this study are available on the request from the corresponding author. The data are not publicly available due to privacy or ethical restrictions.

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
