# Peer review of "A Method to Reduce the Occurrence of Egg Translucency and Its Effect on Bacterial Invasion"

_foods, 2023, doi:10.3390/foods12132538_

Round 1

Reviewer 1 Report

General points about the manuscript: The manuscript brings interesting information regarding the effect of mono-dicalcium phosphate addition to the diet of laying hens on the eggshell translucency when the animals are under heat stress. Authors also investigated whether eggshell translucency affects the cross-eggshell invasion of Escherichia coli. The manuscript is very well written, with sufficient information in the different sections. However, I kindly pointed in this letter some issues that must be reviewed and fixed by the authors, or at least they should explain them better.

Specific considerations:

L20: …August as the…

L26: non-translucent area and non-translucent area?

L28: would not.

L38: add space before the reference.

L57: add scientific names in italic. Please check throughout the manuscript.

L58 and L59: insert space before the reference. Please check throughout the manuscript.

L66-L68: Insert the numbers subscript when appropriate.

L96: Here it says 216 laying hens, but in Abstract is says 72. Please check and fix accordingly.

Table 1: The ingredient content does not sum up to 100. Please precisely indicate the diet composition in a way that all ingredients sum up to 100.

L156: Include the utilized abbreviations in the title of the Figure.

L196: Results – There are letters to compare the results. Is it Tukey test? Please include in Material and Methods. Months are also compared, meaning that months are a factor of the experimental design; please make this idea clear in the Material and Methods, especially in the statistics section.

L197 to L199: Should it be deleted?

L216: Do not repeat in the text those numbers already mentioned in the Table. Please check throughout the manuscript.

L226: Sentence requires revision.

Table 2: Maybe you can clearly state in the footnote that you have mean ±variation. The same for other Tables.

L231. “when feeding was started” A revision of this sentence would make it more understandable.

L241: MSW and MSA.

L249: It’s missing a bracket.

Table 3: Make sure that all abbreviations in the Table are clearly described in footnotes. Check throughout.

L287 to L290: Should it be deleted?

L363: Add space after reference. Please check this issue regarding space throughout the manuscript.

Conclusion very well describes the findings of this study.

L447: Delete dash in “corresponding”. Check L453 as well.

Follow Journal’s guideline for References.

Kind regards.

Only minor editing of English language is required.

Author Response

Response to Reviewer 3 Comments

Dear reviewers:

Thank you for your comments concerning our manuscript entitled “A method to reduce the occurrence of egg translucency and its effect on bacterial invasion” (ID: foods-2462640). Those comments are all valuable and very helpful for revising and improving our paper, as well as the important guiding significance to our researcher. We have studied comments carefully and have made a correction which we hope meet with approval. Revised parts are yellow marked on the manuscript. The main correction in this manuscript and the responses to reviewer’s comments are below:

Point 1: L20: …August as the…

Response 1: Thank you for your advice. We have modified according to your comments. Please see line 20.

Point 2: L26: non-translucent area and non-translucent area?

Response 2: Thank you for your advice. we have modified. Please see line 25-26.

Point 3: L28: would not.

Response 3: Thank you for your advice. We have modified according to your comments. Please see line 27.

Point 4: L38: add space before the reference.

Response 4: Thank you for your advice. We've added spaces before quotes. Please see line 38.

Point 5: L57: add scientific names in italic. Please check throughout the manuscript.

Response 5: Thank you for your advice. We have added scientific names in italics, and the entire manuscript has also been reviewed and italics added. Please see line 56.

Point 6: L58 and L59: insert space before the reference. Please check throughout the manuscript.

Response 6: Thank you for your advice. We've added spaces before quotes. Please see line 57-58. Additionally, we have checked the entire manuscript.

Point 7: L66-L68: Insert the numbers subscript when appropriate.

Response 7: Thank you for your advice. We have added numbers subscript where appropriate in these formulas. Please see line 65-66.

Point 8: L96: Here it says 216 laying hens, but in Abstract is says 72. Please check and fix accordingly.

Response 8: Thank you for your advice. We have checked and it is 72 chickens. Please see line 94.

Point 9: Table 1: The ingredient content does not sum up to 100. Please precisely indicate the diet composition in a way that all ingredients sum up to 100.

Response 9: Thank you for your advice. We checked it, it was a data input error during the tabulation process, and it has been changed. Please see Table 1.

Point 10: L156: Include the utilized abbreviations in the title of the Figure.

Response 10: Thank you for your advice. We have described it in the Materials and methods, in the upper paragraph of Figure 2. Please see line 152-153. Also, in the Results, we have added the abbreviation in the footnotes of all figures and figures.

Point 11: L196: Results – There are letters to compare the results. Is it Tukey test? Please include in Material and Methods. Months are also compared, meaning that months are a factor of the experimental design; please make this idea clear in the Material and Methods, especially in the statistics section.

Response 11: Thank you for your advice. It is not Tukey test. In the experimental design, the month is a part of the experimental design, but only one factor. Because translucent eggs are prone to appear in high temperature and high humidity environment in summer. So we chose to carry out experiments in June, July, August, September, and October in order to increase the number of our experimental samples and verify our conjecture. In Materials and methods for data analysis, we have added specific statistical methods. See line number 155. In the statistical part, we used one-way analysis of variance to process the data, but did not use the Tukey test. We'll take your suggestion into consideration in future experiments. Thanks again for your valuable comments.

Point 12: L197 to L199: Should it be deleted?

Response 12: Yes, we have removed it. Thank you for your careful review.

Point 13: L216: Do not repeat in the text those numbers already mentioned in the Table. Please check throughout the manuscript.

Response 13: Thank you for your advice. We have removed this duplicate data. We also checked the entire manuscript. Please see line 216-222.

Point 14: L226: Sentence requires revision.

Response 14: Thank you for your advice. We have revised the sentence to make it more precise. Please see line 226.

Point 15: Table 2: Maybe you can clearly state in the footnote that you have mean ±variation. The same for other Tables.

Response 15: Thank you for your advice. In view of other reviewers' comments, we changed the presentation of the data to mean ± SEM . In addition, We have marked in the statistical analyses, the basic descriptive statistics of the data are shown as the mean and standard error (mean ± SEM). Please see line 192-193.

Point 16: L231. “when feeding was started” A revision of this sentence would make it more understandable.

Response 16: Thank you for your advice. We've made changes as you suggested. Please see line 232.

Point 17: L241: MSW and MSA.

Response 17: Thank you for your advice. We've made changes. Please see line 243.

Point 18: L249: It’s missing a bracket.

Response 18: Thank you for your advice. We've added a bracket in. Please see line 253.

Point 19: Table 3: Make sure that all abbreviations in the Table are clearly described in footnotes. Check throughout.

Response 19: Thank you for your advice. We have added abbreviations in footnotes to all figures and the manuscript has been fully checked.

Point 20: L287 to L290: Should it be deleted?

Response 20: Yes, we have removed it. Thank you for your careful review.

Point 21: L363: Add space after reference. Please check this issue regarding space throughout the manuscript.

Response 21: Thank you for your advice. We have added spaces after citations and corrected space issues throughout the manuscript.

Point 22: Conclusion very well describes the findings of this study.

Response 22: Thank you for your comment and recognition.

Point 23: L447: Delete dash in “corresponding”. Check L453 as well.

Response 23: Thank you for your advice. We have removed the corresponding dash. Please see line 448, 454.

Reviewer 2 Report

This is generally a well-written paper with only minor revision required.  I have attached the file with highlighted areas accompanied by my comments.

Good English with only minor suggestions.

Author Response

Response to Reviewer 1 Comments

Dear reviewers:

Thank you for your comments concerning our manuscript entitled “A method to reduce the occurrence of egg translucency and its effect on bacterial invasion” (ID: foods-2462640). Those comments are all valuable and very helpful for revising and improving our paper, as well as the important guiding significance to our researcher. We have studied comments carefully and have made a correction which we hope meet with approval. Revised parts are yellow marked on the manuscript. The main correction in this manuscript and the responses to reviewer’s comments are below:

Point 1: , or dicalcium phosphate (DCP)... To be consistent with MCP or MDCP

Response 1: Thank you for your advice. We have modified according to your advice. Please see line 65.

Point 2: Is this in all animals?  Ruminants are commonly supplemented with DCP and they utilize this well due to rumination.  Maybe be more specific and change "animals" to "poultry and swine"?

Response 2: Thank you for your advice. We have changed “animals” to “poultry and swine”. Please see line 69.

Point 3: Is this the correct reference for this statement?  Looks as if it should be reference 21 - maybe just re-check?

Response 3: Thank you for your comments. We rechecked the references and found that we got references 20 and 21 in the wrong order. We have transposed the original references 20 and 21. Please see line 491-495.

Response 4: “powder shell laying hens”, Is this the correct term?

Response 4: Thank you for your comments. We have added the full breed name. Please see line 82.

Point 5: This is redundant since the period is mentioned in the next sentence from July 1 to October 31.  It can therefore be removed or combine this sentence and the one that follows into a single sentence.

Response 5: Thank you for your advice. We have removed this sentence.

Point 6: Not sure what this refers to - is it hens that lay pinkish eggs or a "breed".  If it's a breed one requires something more formal otherwise be more specific.

Response 6: Thank you for your advice. We have added the full breed name. Please see line 95.

Point 7: Maybe this should have the DCP acronym since you have already defined this?

Response 7: Thank you for your comments. We have changed “calcium hydrogen phosphate” to “DCP”. Please see line 98.

Point 8: I assume these are total rather than digestible?  Please elucidate.

Response 8: Thank you for your comments. This is the total methionine content, not the digestible methionine content. We have corrected the description in the table. Please see line 105-106.

Point 9: In addition, blood was collected...  Please remove the word "we". Words such as "we" and "our" may be used in the discussion only where opinioin, speculation or conclusions are noted.

Response 9: Thank you for your advice. We have changed this sentence to”In addition, the blood of the hens was collected every two months.” Please see line 117-118.

Point 10: Please re-word to: Blood biochemical indices were measured every two months,.....  For scientific writing, it is best to remove words such as "we", "I" and other first-person colloquial terms.

Response 10: Thank you for your advice. We have modified the sentence according to your advice. Please see line 122.

Point 11: This is an uncommon term and is a combination of  egg quality: albumen height (Haugh units) and yolk color.  It would be a good idea to define what this unit is and how it is calculated to help inform the reader.

Response 11: Thank you for your advice.

Haugh unit is a very important index to evaluate the quality of eggs. It is an index for checking the freshness of eggs calculated according to the formula by measuring the height of concentrated protein and the quality of eggs. Break the egg and place the contents on a glass plate, keep the egg white layer and egg yolk intact, avoid the frenulum, measure the height of three equidistant points in the center of the egg white layer around the egg yolk, and take the average value as the height of the egg white. Then calculate according to the formula: H.U=100*log (H-1.7*m0.37+7.6), where H is the height of concentrated protein (mm), and m is the mass of eggs (g).

Egg yolk color is the value measured by the multi-function egg tester according to the value of the color fan, and the range is 1.0 ~ 15.0.

The Haugh units and yolk color are all automatically determined by the EMT-5200 multi-function egg tester (Robotmation, Tokyo, Japan).

The multi-function egg tester is an instrument that tests egg weight, albumen height, yolk color, yolk height and egg quality according to the American grade, and uses one device to test the above multiple contents.

These indicators are shown in some literatures, such as literature:

De Juan A F, Scappaticcio R, Aguirre L, et al. Influence of the calcium and nutrient content of the pre-lay diet on egg production, egg quality, and tibiae mineralization of brown-egg laying hens from 16 to 63 wks of age[J]. Poultry Science, 2023: 102491.

Lu, Z., Zeng, N., Jiang, S., Wang, X., Yan, H., & Gao, C. Dietary replacement of soybean meal by fermented feedstuffs for aged laying hens: effects on laying performance, egg quality, nutrient digestibility, intestinal health, follicle development, and biological parameters in a long-term feeding period[J]. Poultry science, 2023, 102478.

Point 12: What unit of measure is YC?  One assumes this is a scale something like the DSM or BASF color fan?

Response 12: Thank you for your comments. Egg yolk color is the value measured by the multi-function egg tester according to the value of the color fan, and the range is 1.0 ~ 15.0. Because the value of the colorimetric fan is a simple division according to the shade of the color, there is no unit of measurement.

Point 13: Your scoring is 0 to 5 - Is this eggs greater than or equal to 3?

Response 13: Thank you for your comments. This grade 3 means that we selected eggs with the translucent egg grade of 3 as the test object when we carried out the E. coli invasion experiment. In order to express what we mean more clearly, we have changed “grade 3” to “the TEG of eggs are 3”. Please see line 168.

Point 14: Is this between 0 and 3 on your grading?

Response 14: Thank you for your comments. This grade 0 means that we selected eggs with the translucent egg grade of 0 as the test object when we carried out the E. coli invasion experiment. In order to express what we mean more clearly, we have changed “grade 0” to “the TEG of eggs are 0”. Please see line 168.

Point 15: Not consistent with the grading in Figure 1?

Response 15: Thank you for your comments. This is based on the grading standard in Figure 1. We selected eggs graded 0 and 3 to do the bacterial invasion test.

Point 16: Re-word to "The experimental culture was inoculated with..."

Response 16: Thank you for your advice. We have changed this sentence to “The experimental culture was inoculated with culture at a dilution of 1:50, and placed it into an ice bath to chill when the culture reached an OD600 of approximately 0.4.” Please see line 173-174.

Point 17: Change "by" to "into"

Response 17: Thank you for your advice. We have deleted this passage.

Point 18: The word "significantly" is not required in these sentences - it is inferred from the P value.  I have highlighted other places where the word is used and could be deleted.

Response 18: Thank you for your advice. We have deleted according to your request.

Point 19: Maybe change this to "by T3, T4 and CORT"...

Response 19: Thank you for your advice. We have made changes according to your advice. Please see line 225.

Point 20: Change to "common items" rather than "the same".

Response 20: Thank you for your advice. We have made changes according to your advice. Please see line 258.

Point 21: What unit of measure is this?  It hasn't been clearly defined and nor has the HU value.

Response 21: Thank you for your comments. Yolk color (YC) is the value measured by the multi-function egg tester according to the value of the color fan, and the range is 1.0 ~ 15.0. Because the value of the colorimetric fan is a simple division according to the shade of the color, there is no unit of measurement.Aad Haugh unit (HU) is a very important index to evaluate the quality of eggs. It is an index for checking the freshness of eggs calculated according to the formula by measuring the height of concentrated protein and the quality of eggs.

Point 22: Change this to "common items" as well.

Response 22: Thank you for your advice. We have made changes according to your advice. Please see line 267.

Reviewer 3 Report

see attachment 

Author Response

Response to Reviewer 2 Comments

Dear reviewers:

Thank you for your comments concerning our manuscript entitled “A method to reduce the occurrence of egg translucency and its effect on bacterial invasion” (ID: foods-2462640). Those comments are all valuable and very helpful for revising and improving our paper, as well as the important guiding significance to our researcher. We have studied comments carefully and have made a correction which we hope meet with approval. Revised parts are yellow marked on the manuscript. The main correction in this manuscript and the responses to reviewer’s comments are below:

Point 1: Changes to article titles

Response 1: Thank you for your advice. We have changed the title to “A method to reduce the occurrence of egg translucency and its effect on bacterial invasion” as you suggested. Please see line 1-3.

Point 2: Translucent eggs consumption is low due to consumer acceptance and quality concerns, which is a problem that egg producers need to address. Whether to add be after need to in this sentence

Response 2: Thank you for your advice. After checking again, we found that this sentence pattern does not need to be modified. This sentence is to express that the problem is that egg manufacturers need to solve the problem of translucent eggs.

Point 3: strain? average body weight and SD

Response 3: Thank you for your advice. The mean body weight and SD of the hens are described in Materials Methods. Please see line 95.

Point 4: average temperature and humidity

Response 4: Thank you for your advice. Temperature and humidity are shown in result 3.1.

Point 5: On the Problem of P-Value Representation of Data in the Full Text.

Response 5: Thank you for your advice. We have made corrections according to your suggestion.

Point 6: Summarily, our stuies shows that dietary MDCP may alleviate eggshell translucency and that eggshell translucency wouldn’t increase the probability of E. coli cross-shell penetration rate.

Response 6: Thank you for your advice. We have changed “Summarily” to “in conclusion” and “studies” to “results” in this sentence as you suggested. Please see line 26.

Point 7: Egg consumption is focused mainly on shell eggs.

Response 7: Thank you for your advice. We have deleted this sentence according to your suggestion.

Point 8: food

Response 8: Thank you for your advice. We have changed “food” to “feed”. Please see line 44.

Point 9: which experimental design was followed?

Response 9: Thank you for your comments. In the grouping process, in order to reduce the error, we adopted the experimental design of random grouping of hens. In order to express more accurately, we have modified the statement. Please see line 95-96.

Point 10: add mathematical model for better illustration of data analysis

Response 10: Thank you for your advice. We have added mathematical models and descriptions. Please see line 186-191.

Point 11: add graph of Thermal Head Index

Response 11: The initial design of our experiment was to explore the effect of adding MDCP in the feed on the translucent shape of eggs, and we only used a thermo-hygrometer to simply record the temperature and humidity. Therefore, the exact value of THI cannot be calculated. However, in a follow-up experiment, we measured THI for the temperature and humidity in the same house (as shown in the figure below). Considering the rigor of the data, we did not use this batch of data. Thank you again for your suggestion, tand the follow-up experiment will be more rigorously designed.

Point 12:company details Model and country

Response 12:Thank you for your advice. We have added the model, specification and country of the LED. Please see line 133.

Point 13: author's should add Egg Shell Pores Number (estimate):Check Rahn and Paganelli, 1990

Response 13:Thank you for your advice. We have previously investigated the correlation between egg shell pores number and egg translucence shape, but there was no significant correlation between the two. this research has been published(Shi, X.; Li, x.;Xiong, M.; Lin, Y.; Zeng, L.; Song, j.; Liang, Q.; Xu, G.; Zheng, J. Study on the Effect of Translucency on Bacterial Penetration in Eggs and Egg Quality. Chinese Journal of Animal Husbandry. 2022,58(12):277-280. (In Chinese). https://doi.org/10.19556/j.0258-7033.20211123-06.). Therefore, in this experiment, we did not record and investigate the relationship between the number of eggshell stomata and the translucent shape of the egg.

Point 14: Hastelloy units

Response 14:Thank you for your advice. We have changed “Hastelloy units” to “Haugh unit”. Please see line 159.

Point 15: why authors use SD ? SEM is better and reliable estimate  add SEM

Response 14:Thank you for your advice. We have added SEM.

Point 16: results:add actual p-value instead of relative

Response 16: Thank you for your advice. we have added actual p-value instead of relative.

Point 17: what is this?

Response 17: Thank you for your advice. we have removed this sentence.

Point 18: in addition to this  add line graph of THI during experimentation period

Response 18: Thank you for your advice. To solve this problem, see the answer to point 11。
